



# A BRDF-BPDF database for the analysis of Earth targets reflectances

Francois-Marie Breon[1], Fabienne Maignan[1]

[1]Laboratoire des Sciences du Climat et de l'Environnement (LSCE/IPSL), UMR CEA-CNRS-UVSQ,
University Paris-Saclay, Gif-sur-Yvette, 91191, France

*Correspondence to*: Francois-Marie Breon (fmbreon@cea.fr)

To be submitted to Earth System Science Data http://earth-system-science-data.net/

**Abstract.** Land surface reflectance is not isotropic. It varies with the observation geometry that is defined by the sun and view zenith angles and the relative azimuth. In addition, the reflectance is
linearly polarized. The reflectance anisotropy is quantified by the Bidirectional Reflectance Distribution Function (BRDF) while its polarization properties are defined by the Bidirectional Polarization Distribution Function (BPDF). The POLDER radiometer that flew onboard the PARASOL micro-Satellite remains the only space instrument that measured numerous samples of the BRDF and BPDF of Earth targets.

Here, we describe a database of representative BRDF and BPDF derived from the POLDER measurements. From the huge amount of data acquired by the spaceborne instrument over a period of 7 years, we selected a set of targets with high quality observations. The selection aimed at a large number of observations, free of significant cloud or aerosol contamination, acquired in diverse observation geometries with a focus on the backscatter direction that shows the specific Hot-Spot signature. The
targets are sorted according to the 16-classes IGBP land cover classification system and the target selection aims at a spatial representativeness within the class. The database thus provides a set of high quality BRDF and BPDF samples that can be used to assess the typical variability of natural surface reflectances or to evaluate models. It is available freely from the PANGAEA website (https://doi.pangaea.de/10.1594/PANGAEA.864090).

In addition to the database, we provide a visualization and analysis tool based on the IDL language. It allows an interactive analysis of the measurements and a comparison against various BRDF and BPDF



analytical models. The present paper describes the input data, the selection principles, the database format and the analysis tool.

## 1    Introduction

The albedo of a target is the fraction of the incoming light that is reflected rather than absorbed by the
surface (Schaepman-Strub et al., 2006). It varies between 0 (full absorption) and 1 (full reflection). The albedo of natural Earth targets varies widely depending on the surface types: Vegetation absorbs most of the incoming visible light whereas the opposite is true for snow. In addition, the albedo varies with wavelength. Many land surface characteristics can be inferred from the spectral signature of their albedo. Spectral indices such as the Normalized Difference Vegetation Index (NDVI) have been
developed to quantify the amount and state of vegetation or other properties (Carlson and Ripley, 1997;Asrar et al., 1984).

The albedo is a quantity that integrates the reflected light over all directions of the hemisphere. This quantity is difficult to measure as a typical radiometer measures the reflected light in a single direction. This is particularly true for spaceborne observations where a target is observed from a given direction.
As a direct consequence, the radiometer is not sensitive to the Albedo but rather to the reflectance (Schaepman-Strub et al., 2006).

Global observation from space and multi-temporal monitoring of a given target impose the combination of measurements acquired with different observation geometries (Lunetta et al., 2006). It is well known that the reflectance of natural surfaces is far from isotropic (Bicheron and Leroy, 2000). For most land
surfaces, the reflectance is larger in the backscatter direction than it is in the forward direction. In the very few degrees towards the backscatter direction, the reflectance increases markedly, an optic phenomenon referred to as the Hot Spot (Breon et al., 2002), but that is difficult to quantify properly at the measurement is perturbed by the radiometer own shadow. Snow is much more isotropic than other surfaces, but nevertheless shows larger reflectance values in the forward than in the backward
hemisphere, which is opposite to that of other land surfaces (Peltoniemi et al., 2005).

The directional signature of the reflectance is described by the Bidirectional Reflectance Distribution Function (Schaepman-Strub et al., 2006). In principle, it is a function of four angles, the illumination

(solar) and view zenith and azimuth angles. In practice, and except for targets that show a preferential direction, such as crops planted along rows, the azimuths are only significant by their difference. Thus, the BRDF is most often described as a function of $(\theta_s, \theta_v, \phi)$ where $\theta_s$ and $\theta_v$ are the solar and view zenith angles and $\phi$ is the relative azimuth.

The goal of this paper is to describe a database of BRDF samples that has been developed based on spaceborne measurements of the Earth reflectances. This database may be used to assess the variability of land surface BRDF, for the development and validation of BRDF models, and as a boundary condition for atmospheric radiative transfer studies.

Other characteristics of the land surface reflectance are its polarization properties. The incoming direct
solar light is unpolarized. Conversely, the light scattered in the atmosphere by molecules and aerosols, and the light reflected by the surface is partly polarized. Few optical instruments designed to monitor the Earth have polarization capabilities and much less efforts have been devoted to the polarization characterization of land surfaces than to the BRDF. However, polarization is a great tool to monitor anthropogenic aerosols and clouds from space, as demonstrated with the POLDER instrument (Waquet
et al., 2009b;Deuze et al., 2001;Breon and Doutriaux-Boucher, 2005). This led to the development of the Glory mission (Mishchenko et al., 2007) that was unfortunately lost at launch. The 3MI instrument, which is similar to POLDER but with advanced capabilities in terms of spatial resolution and spectral coverage, shall be onboard the forthcoming series of Eumetsat MetOp satellites (Marbach et al., 2015). A primary objective of this space mission is the monitoring of atmospheric aerosols and clouds using
the polarization characteristics of the reflected light.

Information about the land surface polarization characteristics is therefore needed. The database that is presented in this paper includes, in addition to the spectral reflectances, the polarization characteristics in one channel.

In the following, we describe the input data, the data processing and selection and the database format.
In addition, we have developed an interactive tool to allow a simple graphical analysis of the database and a confrontation to analytical models. The tool is therefore described in the second part of the paper with a few examples of its outputs.



## 2 Input data and processing

### 2.1 The POLDER instrument onboard the PARASOL mission

The POLDER-1 and POLDER-2 radiometers have been onboard the ADEOS 1 and 2 platforms in 1996-1997 and 2003 respectively (Deschamps et al., 1994). Unfortunately, the solar panel of both

satellites failed after a few months of operations so that only 8 and 7 months of measurements were available from these instruments. This limitation did not allow the monitoring of a full vegetation cycle, which strongly reduced the interest in the data. Fortunately, a new opportunity occurred with the development by CNES of a line of micro-satellite platforms. The POLDER instrument was selected to be installed onboard one of these platforms and became a member of the A-Train to complement the

other instruments. The satellite was name PARASOL after *Polarization and Anisotropy of Reflectances for Atmospheric Sciences coupled with Observations from a Lidar* (Tanre et al., 2011).

The experience gained with POLDER-1 and 2 was used and resulted in a few changes on the instrument, in particular regarding the choice of the spectral bands. There are eight spectral bands for the POLDER/PARASOL instrument with central wavelengths from 443 nm to 1020 nm. One main

feature of POLDER is its capability to measure the linear polarization of the light in three channels centred at 490, 670 and 865 nm. This is achieved through three successive measurements with identical spectral filters and three polarizers rotated by step of 60°. The processing of these measurements provides the radiance intensity, its polarization degree and the polarization direction or, alternatively, the Stokes vector components ($I$, $Q$ and $U$).

The other main specificity of the POLDER instrument is its ability to provide multi-directional measurements. This is possible thanks to its optical design that consists of a wide field of view lens associated with a bi-dimensional CCD matrix. This combination generates a bi-dimensional Field of View with forward/backward angles of ±51° and crosstrack angles of ±43°. The maximum view angle at the surface is close to 70°, and corresponds to measurements acquired around the corners of the CCD

matrix. As the satellite flies over, up to 16 (average is 14) observations of the target are available. These observations provide a sampling of the target BRDF. During the following days, the PARASOL satellite flies again over that target, albeit on a different orbit, which provides another set of BRDF samples. Depending on cloud cover, these successive measurements allow a very broad sampling of the





BRDF for view angles up to ≈60°, assuming a stability of the target during the composition period. Note however that PARASOL is on a helio-synchronous orbit so that the various acquisitions are made at a near constant solar time. As a consequence, there is little variation of the sun angle in the measurements of the target reflectance during a short period.

The Parasol satellite was launched in December 2004. Data acquisition started in early 2005 and was nearly continuous until October 2013. However, due to lack of fuel, the satellite left the A-Train and was on a slowly drifting orbit after December 2009. There have been some data acquisition interruptions during the lifetime of the satellite, mostly resulting from malfunctions of the stellar sensor.

The best year in term of data acquisition was 2008. As a consequence, we selected that year to build the BRDF/BPDF database.

After the end of the on-orbit operations, POLDER/Parasol data benefited from further development in the calibration and data processing. Using several vicarious calibration techniques, all based on natural targets, it was possible to derive an accurate set of calibration parameters that account for the temporal

evolution of the instrument sensitivity characterized by a mean decrease modulated by a variation within the field-of-view (Fougnie, 2016). These progresses led to a full reprocessing of the POLDER/Parasol dataset at the end of 2015.

## 2.2  POLDER data processing

The POLDER instrument provides Top of the atmosphere Reflectances after calibration (Fougnie et al.,

2007). These Level-1 measurements are processed into Level-2 products using several processing chains. The reflectances are corrected for atmospheric absorption ($H_2O$, $O_3$, $O_2$, $NO_2$). Over land, the atmospheric aerosol load is estimated from the polarized reflectance measurements using pre-computed tables (Deuze et al., 2001). The reflectance measurements are then corrected for atmospheric scattering for an estimate of the spectral surface reflectance. The polarized reflectances are corrected for the

molecular scattering; they are *not* corrected for aerosol scattering.

The so-called Level-2-A official product contains an estimate of the directional surface reflectance for 6 spectral bands, and an estimate of the directional surface polarized reflectance at 865 nm. Only the



longer wavelength channel is provided as (i) it is generally assumed (and the POLDER aerosol inversion does so) that the surface polarized reflectance is spectrally neutral (Waquet et al., 2009a) and (ii) the atmospheric contribution is dominant and more difficult to correct for the shorter wavelength channels. The product also includes a non-quantitative indication of the aerosol load.

Some explanation is needed on what we refer to as the "polarized reflectance". As said above, the POLDER instrument measures the Stokes vector representation [*I,Q,U*] of the radiance. A reference plane is needed to define $Q$ and $U$. Many studies use the vertical plane (that contains the view and local nadir directions) as a reference. Yet, it is more practical to use the scattering plane (that contains the sun and view direction). With this plane as a reference, $U$ is most often very small with respect to both

$I$ and $Q$ (Schutgens et al., 2004). This is because the polarization is either parallel or perpendicular to the plane of scattering. $Q$ is smaller than $I$ but takes measurable values. In most cases, the polarization is perpendicular to the plane of scattering so that $Q$ is negative. In rare cases, the polarisation is parallel to the plane of scattering in which case $Q$ is positive. We thus define the polarized reflectance $R_p$ as:

$$R_p = \frac{-\pi Q}{E_0 cos\theta_s} \tag{1}$$

in a similar way as the reflectance definition:

$$R = \frac{\pi I}{E_0 cos\theta_s} \tag{2}$$

where $E_0$ is the TOA solar irradiance. With such definition, $R_p$ is most often positive, but it nevertheless contains the information whether polarization is perpendicular or parallel to the plane of scattering.

## 2.3  Data selection

The objective is to sample the variability of land surface BRDF and BPDF while selecting only the observations that are free from significant aerosol and cloud contamination and for which a large number of observations were available.

Early studies have shown that there are systematic changes of the BRDF with the land surface type (Bacour and Breon, 2005):-. As mentioned above, snow has a very specific directional signature; desert

show a more isotropic directional signature than vegetated surfaces, wetlands sometime show a glint




signature in addition to the classical maximum in the backscatter direction... It is then natural to sample BRDFs as a function of the land surface cover. For this objective, we make use of the IGBP classification (Loveland et al., 1999). We used the official MODIS land cover product (MCD12Q1) for the year 2008 at 5 minute resolution (Liang et al., 2015). For each POLDER pixel ($\approx$6.2x6.2 km$^2$) we

analyse the land cover type for the 5x5 MODIS cells centred on the POLDER pixel. Only the POLDER pixels for which there is a clear dominance of one land cover type (>75%) are kept for further processing. The POLDER pixels are assigned the IGBP land cover type as identified from the MODIS product and the relative fraction of the dominant type is kept for inclusion as ancillary information in the database.

For each POLDER pixel that passes this first step, and for each of the 12 months independently, we retrieve all POLDER/Parasol directional observations that pass the cloud detection scheme. A BRDF model is fitted against the 670 nm surface reflectances and the Root Mean Squared Error (RMSE) between measurements and model is computed. The objective is to reject poorly corrected aerosol contamination, that increases the RMSE, and keep pixels with a large number of observation.

A score for the pixel-month is defined as:

$$score[p, m] = \sqrt{Nmes}/RMSE \tag{3}$$

where $p$ identifies the POLDER pixel and $m$ identifies the month; Nmes is the number of directional POLDER measurements that are available. In addition, as there is a particular interest in the analysis of the Hot-Spot directional signature, we increase the score by 20% if the set of directional measurements

includes at least one with a phase angle of less than 1°.

We also compute a yearly score as the sum of the monthly scores:

$$scoreY[p] = \sum_m score[p, m] \tag{2}$$

For each IGBP surface type and each month, we select the 50 "best" targets, i.e. those that have the highest score. On the other hand, we seek some diversity and thus want to avoid selecting pixels that

are close to one another. We therefore select pixels iteratively: After a pixel with the highest score is selected, that of all pixels is multiplied by $\left(1 - exp(d/100)\right)$ where $d$ is the distance (in km) between





each of these pixels and that selected at the previous step. The score of the nearby pixels is then reduced which insures that they are not selected subsequently.

As a result of this procedure, we select independently 50 targets for each of the 12 months and each of the 16 IGBP surface types. This procedure leads to the *monthly database*.

In addition, we generate a *yearly database* where the selection is based on the yearly score *scoreY* rather than the monthly scores. The procedure is very similar. In the *yearly database*, the same targets are selected for the 12 months. Conversely, the *monthly database* selects pixels independently for each month which results, in most cases, in different target sets. The monthly database is best to analyse

targets of high quality for each month independently. The yearly database shall be used to assess the variability of the BRDF and BPDF along the year, as shown in section 3.7 below, although some months may be poorly sampled.

### 2.4 Database structure

The two databases (*monthly* and *yearly*) are built around a large number of text files (≈16x12x50). Each

file includes the surface reflectance and polarized reflectance acquired during the month. The files are sorted by IGBP surface types (`nn` from 01 to 16) and then by month (`mm` from 01 to 12): The directory `IGBP_nn` contains the subdirectories `2008mm` which contain the files. The file format is described in Appendix A.

In addition, the database includes a binary file `map_IGBP.bin`. It reproduces the IGBP classification

used for the data selection on a 540 x 270 (lon x lat) grid. This file is used by the graphic analysis tool.

### 3 Database's features as evidenced by the analysis tool

### 3.1 Visu_brdf tool set up

A graphical interface tool has been developed to analyse the BRDF/BPDF data file described above. The code `visu_brdf.pro` is based on the IDL language and its use requires an IDL licence. Another

option, that does not require an IDL licence, is to download the *IDL Virtual Machine* from the *Harris*



*Geospatial* web site (https://www.harris.com/what-we-do/geospatial-solutions). The virtual machine lets you run the compiled version of the analysis tool, provided in the `visu_brdf.sav` file.

The first step is to locate the database. Inside the code, you shall change the variable "`HomePath`" to the directory that contains the monthly and yearly databases.

When using IDL with a proper licence, one shall type:

```
IDL > .compile visu_brdf.pro
IDL > visu_brdf
```

If the "`HomePath`" was not set properly, a warning message indicates that one must select the path for the "POLDER BRDF" database and a window opens up for that purpose.

If one uses the *IDL virtual Machine* option, just double-click on the `visu_brdf.sav` icon. As described above, one must then select the path for the database.

### 3.2 The *Main Command Window*

Figure 1 shows the *Main Command Window* of the BRDFs analysis tool. In the following, it is referred to as *MCW*. One can select one of the IGBP surface type with the "`BIOME`" drop-down list or successive clicks on the "`NEXT`" button that is next to it. Similarly, one can select the time period with the "`Month`" drop-down list, and the NDVI range with the "`NDVI`" drop-down list.

The available targets are shown on the map. Note the lighter grey areas that indicate the Earth surface that corresponds to the selected IGBP surface type. The squares indicate the locations of the targets in the BRDF database that correspond to the criteria (IGBP type, month and NDVI range). The colour of the squares is either from black to red (we use a rainbow palette) according to the NDVI if "`ALL NDVI`" is selected or red if a specific range of NDVI is selected.

The selected target is indicated by a red circle. There are several ways to choose a target for display of its measurements. The easiest one is to click on the map close to the desired square. The second is to use the "`Next_BRDF`" button that selects the various targets in successive order. Finally, it is also possible to use the "`Random`" button that selects a target randomly among the ones shown on the map.



Below the map are given some information on the selected target: *latitude* and *longitude*, *surface type*, *fraction of this surface type*, the *number of orbits* (satellite overpasses) and the *total number of observations* for this target.

When a target is selected, several windows are displayed and can be changed with various options.

They will be described below.

Below the BRDF and BPDF models selection is a box with check buttons that change the measurement model visualisation. The `Meas / Surf / Isoc` checkboxes affect the "*Target BRDF*" window and are described below. The "`Polariz`" checkbox controls the display, or not, of a specific window for

the BPDF. The "`PPlog`" checkbox toggles between a linear and log scale for the "*Principal plane*" and "*Perpendicular plane*" windows that are described below.

Further below, the checkboxes with the band central wavelengths make it possible to select three bands, from the 6 provided in the database, for display in the "*Target BRDF*", "*Principal Plane*" and "*Perpendicular Plane*" windows.

Finally, the different buttons of the *Main Command Window* located below the information on the target allow to save a measurement-model comparison in various formats:

"`PNG`" saves the window in PNG format,

"`Poscript`" generates a poscript file,

"`Encaps.`" generates encapsulated Poscript file (`file.eps`). Note that Figure 2 to Figure 8 have all been generated using that method.

The "`SaveWindow`" button makes a simple copy of the current window into another one. This is useful to compare the predictions of different models.

The "`Clear`" button erases all such windows.





### 3.3 BRDF/BPDF models within visu_brdf

The *visu_brdf* tool can be used to compare the BRDF and/or BPDF measurements to analytical models. Several such models have been implemented within the tool and can be selected by the two drop-down lists "`BRDF model`" and "`BPDF model`".

Currently 7 BRDF models are available in the visu_brdf tool:

- **Ross-Li** (linear, 3 parameters): used in the MODIS processing (Schaaf et al., 2002)
- **Roujean** (linear, 3 parameters): used in the original POLDER processing (Roujean et al., 1992)
- **Ross-Li HotSpot** (linear, 3 parameters): with the Hot Spot modelling (Maignan et al., 2004)
- **Roujean HotSpot** (linear, 3 parameters): with the Hot Spot modelling (Maignan et al., 2004)
- **Engelsen** model (semi-linear, 3 parameters) (Engelsen et al., 1998)
- **RPV** (non-linear, 3 parameters) : model of Raman-Pinty-Verstraete (Verstraete et al., 1990)
- **Snow** (linear, 1 parameter): specific for snow surfaces (Kokhanovsky and Breon, 2012)

Similarly 4 BPDF models are available in the visu_brdf tool:

- **Nadal** (linear, 2 parameters): used in early times of the POLDER experiment (Nadal and Breon, 1999)
- **Breon** (linear, 2 parameters): two kernels developed for vegetation and bare soils (Breon et al., 1995)
- **Maignan** (linear, 1 parameter): developed from POLDER measurements (Maignan et al., 2009)
- **Litvinov** (non-linear, 3 parameters) (Litvinov et al., 2011)

There are plans to develop a BPDF model for snow surfaces which may then become available for further releases of the *visu_brdf* tool.

### 3.4 Analysis of a target BRDF

Figure 2 shows an example of the *Target BRDF* window content. This window shows the reflectance
measurements as well as a comparison against the modelling results, after a best fit against the measurements. The BRDF model that is used is selected on the *MCW*. Each row of the figure





corresponds to one chosen wavelength among the six available wavelengths (490, 565, 670, 765, 865 and 1020 nm) on the bottom-left of the *MCW*.

The reflectance shown in Figure 2 are typical for a surface with some vegetation. The reflectance is significantly larger in the near infrared (865 nm) than it is in the red (670 nm) or green (565 nm). For a

given wavelength, the reflectance increases towards the backscatter direction. The model is able to reproduce most of the directional variation, as shown on the scatter plot (right column), and the model-measurement correlation is more than 0.97. The central column shows the difference between measurement and modelled reflectances. In Figure 2, these differences appear mostly random and do not show a systematic variation within the directional space. This indicates that there is little hope for a

BRDF model that fits the measurement better. Other targets show measurement-model differences with more spatial structure, indicating a deficiency in the modelling that might be improved (not shown).

By default, the left column shows the reflectance measurements as shown on Figure 2. It is also possible to use other displays of the measurements as selected with the toggle buttons on the *MCW*. The option:

• "`Meas`": shows the reflectance measurements (default),

     • "`Isoc`": shows the isolines of the model outputs (after a best fit),

     • "`Surf`": shows the modelled reflectance as a coloured surface. This option disables the two others.

The data are plotted on a polar diagram. The large circles correspond to view zenith angle of 20, 40 and

60°. The principal plane is on the horizontal line with backscatter to the right. The perpendicular plane is along the vertical line. The small circle on the horizontal line corresponds to the median sun angle during the period of synthesis and indicates the backscatter direction.

Note that the reflectance-difference ranges for the colour scales vary with the channel and are indicated on the bottom of each plot. Finally, information on the target are mentioned at the bottom of the figure:

Location, period, IGBP surface type, and analytical model used for comparison.



### 3.5 Analysis of directional signatures in the principal and perpendicular planes

Figure 3 presents an example of directional signatures in the principal and perpendicular planes. Such figures are automatically generated by the *visu_brdf* tool when a target and a model have been selected. The two windows show the measured (triangle symbols) and modelled (lines) reflectances in the

principal and perpendicular planes for the three selected wavelengths as a function of the view zenith angle. Only those measurements close to the principal/perpendicular planes are displayed. The measurements are corrected for the BRDF variations between the geometry of observation and the principal/perpendicular plane, i.e. the symbols indicate:

Measurement + Model(principal/perpendicular plane) – Model(Viewing geometry)

This correlation is small but are necessary to account for the variation of the sun angle within the month, and the small variations of the reflectance between the observation geometry and the parallel/perpendicular plane.

The Y-axis can be either on a linear or log scale depending on the "PPlog" option in the *MCW*. Similarly, the 3 channels that are displayed can be modified among the 6 that are available in the

database. These figures confirm the general observation made over Figure 2: The reflectance is significantly larger in the near-infrared than it is in the visible; it increases markedly from forward scatter towards backscatter, while there is insignificant variations in the perpendicular plane; the model reproduces properly the observed variations.

### 3.6 Analysis of a BPDF target

Figure 4 shows an example of the content of the *Target BPDF* window, which is very similar to the *Target BRDF* except that it shows the surface polarized reflectance at a single wavelength (865 nm). Although the POLDER instrument made polarized measurements in three channels, only the longer wavelength channel is provided in the database and, therefore, accessible through the *visu_brdf* tool. Our experience is that the surface polarized reflectance is spectrally neutral, or that the spectral

variations are smaller than the measurement noise. We thus provide the longer wavelength channel estimates that are the least contaminated by atmospheric scattering.





The left image shows the measurements; the middle image is the model-measurement difference, and the right figure is a scatter plot of the measurement and model. The same ancillary information as for the *target BRDF* window is provided on the bottom of the window.

The directional signature of the polarized reflectance is completely different than for the reflectance. At
backscatter, the polarized reflectance is very small and even negative, indicating a polarization parallel to the plane of scattering. The polarized reflectance tends to increase with the phase angle away from backscatter. Note that the polarized reflectance is much smaller than the reflectance, so that the polarisation ratio is only a few %. Although the model does a fairly good job, it does not reproduce the negative polarization close to backscatter. The scatter plot indicates two different regimes where the
modelling is clearly larger or clearly smaller than the observation. The directional diagram (middle) does not show any systematic feature. Positive and negative differences are observed in very similar observation geometry. This indicates a slight change in the target polarized reflectance within the period of synthesis.

### 3.7 Analysis of the temporal variations of the BRDF parameters using the yearly database

As explained in the database description above, there are in fact two databases. The *Monthly database* processes all months independently and selects the best targets for each month. Conversely, the *Yearly database* selects a set of targets for the full year. With the latter, it is then possible to analyse how the BRDF and BPDF parameters vary along the year. To use the Yearly database with the visu_brdf tool, either use the keyword /YEAR if the HomePath has been properly defined:

```
IDL> visu_brdf,/YEAR
```

Or provide the full path to the *Yearly Database*, e.g.:

```
IDL>visu_brdf,pathin='/home/users/breon/BRDF_database_Year'
```

In such case, the *MCW* is very slightly different than with the Monthly Database as shown in Figure 5. There are three additional checkboxes below the drop-down buttons used for model selection.

These checkboxes, `TimeSeries`, `TSoption` and `All_BRDF` control the display of the windows that are described below. The `TimeSeries` checkbox must be selected to show the *Vegetation Indices window* and the *Time Series window* while `TSoption` controls its content. Similarly, the `All_BRDF` checkbox controls the display of the corresponding window.





### 3.7.1 Vegetation Indices time series

The *Vegetation Indices* window shows the time series of NDVI (in green) and 3*DVI (in blue) over the full year. 3*DVI is shown rather than DVI to get a range similar to that of NDVI. DVI is the simple difference of the 865 nm and 670 nm channels reflectances. NDVI is the normalized difference of these

parameters. The reflectance is the nadir value derived from the selected BRDF model after a fit on the measurements. The purpose of the *Vegetation Indices* window is mostly to provide some indication about the vegetation cover variations within the year for a better interpretation of the figures that are discussed below. On the example shown in Figure 6, there is a clear annual cycle of the vegetation cover with an increase of the vegetation indices during the spring and a dry down during the fall.

### 3.7.2 Model parameters time series

The *Time Series* window displays the annual time series of the three parameters of a linear BRDF model, referred to as $k_0$, $k_1$ and $k_2$. The model used is that selected in the MCW but, in the current version, the *Time Series* window only functions for the linear BRDF models (i.e. not *Engelsen*, *RPV* and *Snow*). If the TS Option checkbox is set, it displays, from top to bottom, the time series of the

reflectance in a particular geometry (sun at 40° from zenith and view at nadir) and the ratio of model parameters: $k_1/k_0$ and $k_2/k_0$. An example is shown on the right side of Figure 7. When the checkbox is not set, the time series of $k_0$, $k_1$ and $k_2$ are shown, as on the left side of Figure 7. The time series of the three selected channels are shown in color while the others channels are also displayed but in black. Note that the parameters are displayed only if the coefficient of correlation between measurement-

model is larger than the "Time Series Min. Corr" threshold set in the *MCW*. Indeed, when the correlation is low, the BRDF coefficients have little value and should not be displayed. The user can change the threshold and see its influence on the results.

The example of the Time Series window shown in Figure 7 indicates that the target BRDF changes with the vegetation growth and decay. Indeed, $k_1$ and $k_2$ (left), or their ratio with $k_0$ (right) vary

concomitantly with the vegetation indices. Although it is not seen for all bands, $k_1$ tends to decrease with an increase of the vegetation index while $k_2$ tends to increase. This behaviour is found over most targets (Breon and Vermote, 2012) and is somewhat expected as $k_2$ is associated with the RossThick





kernel which aims at modelling the BRDF of a thick canopy. Note that the time series are often incomplete because of a lack of observations (because of cloud cover or sun angle issues). On the example of Figure 7, there are no parameter estimates for December.

The time series on this example are relatively clean. There are cases with more variable parameter
retrievals. The blue (490 nm) band over the vegetation is particularly difficult because of the low surface signal and the large atmospheric correction.

### 3.7.3 BRDF/BPDF seasonal evolution

Finally, when the `All_BRDF` checkbox is set, the *All BRDF/BPDF* window shows the BPDF (first line) and BRDF (following lines) for the 12 months. An example is shown in Figure 8. On each of the polar
diagrams that are shown, one can identify the independent satellite overpasses, with up to 16 observation directions that are roughly aligned in the angular space. The general orientation of these observations vary along the year because the sun azimuth, at the local time observation, varies.

All polar plots show the main characteristics that have been described earlier, with a maximum reflectance and a minimum polarized reflectance close to backscatter. Figure 8 also shows a change in
the general reflectance along the year. At 865 nm, the reflectance is the largest in August, when it appears to be the lowest at 670 nm. This observation is fully consistent with the change of the vegetation index (Figure 6) and the BRDF parameters (Figure 7) along the year. The *All BRDF/BPDF* window is appropriate to get a full view of the observation of a given target along the year, while the other windows are required for a more quantitative interpretation.

**4    Conclusions**

The main focus of the POLDER spaceborne instrument was for atmospheric studies, i.e. the monitoring of aerosols and clouds. It may be argued that the spatial resolution of 6x6 km$^2$ is not suitable for the analysis of land surface processes. However, we have here selected homogeneous targets, in which case the spatial resolution of the measurement is not an issue. Although the first version of the
instrument was launched 20 years ago (1996 on the ADEOS-1 platform), it remains the only instrument that measures the full linear polarization of the Earth reflectance in the solar domain. Besides, the





directional coverage of POLDER is better than that of MISR (Diner et al., 1998), the only other instrument that provides multi-directional sampling of the Earth reflectances (Lallart et al., 2008). POLDER remains therefore, an up-to-date tool for the analysis of the directionality and polarization of land surface reflectances. We have developed a database for the remote sensing community that

provides a description of representative Earth targets. A similar undertaking has been achieved based on airborne measurements at a higher spatial resolution (Gatebe and King, 2016). Earlier version of the database had been developed based on the measurements from the POLDER instrument onboard the ADEOS and PARASOL satellites. Although these versions have not been properly described in the peer-reviewed literature, they have been used for several analysis of the surface directional (e.g.

Kokhanovsky and Breon, 2012; Cui et al., 2009; Jiao et al., 2014; Bacour and Breon, 2005; Maignan et al., 2004) and polarization (e.g. Litvinov et al., 2012; Maignan et al., 2009) signatures. The new version, which is described in this paper, is of better quality. It benefits from improved calibration and data selection scheme; it provides the reflectance measurement over an extended spectral range (up to 1020 nm) and it is associated with an interactive analysis tool. These data can be used to develop new

models and evaluate their ability to reproduce the observed spectral, directional and polarization signatures. The database and the analysis tool are available free of charge for the scientific community from the PANGAEA website (https://doi.pangaea.de/10.1594/PANGAEA.864090).





## 5    Acknowledgments

The work that led to this database was made possible thanks to the support from CNES and Eumetsat.
The POLDER/PARASOL Level 1 and Level-2 products have been generated by CNES and are
distributed by ICARE (http://icare.univ-lille1.fr)

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





## 7 Appendices

### 7.1 Appendix A: File structure

Each file contains all PARASOL measurements for a target acquired during a month. The name of the file `brdf_ndvi`**`NN_LLLL_CCCC.`**`txt`" indicates its NDVI range and its location:

**NN** indicates the NDVI range.

**LLLL** is the line number in the POLDER sinusoidal grid (1 to 3240)

**CCCC** is the column number in the POLDER sinusoidal grid (1 to 6480)

The first lines of a BRDF/BPDF data file are given below:

```
latitude  longitude  IGBP_class   NDVI  nb_orbit  nb_dir  homogeneity(%)
       65.47     119.58        3       0.32     15      210        100
    yymmdd Orbit   SZA  VZA RelAzi AziS   DVzC    DVzS    R490  R565  R670  R765  R865 R1020   Rp865 Aero
    080307 075061 70.7 59.2  15.8 186.0 -0.068 -0.041   0.394 0.374 0.372 0.400 0.386 0.378  0.0012  2
    080307 075061 70.7 54.3  14.4 186.0 -0.074 -0.044   0.390 0.367 0.360 0.387 0.381 0.358  0.0005  2
15  080307 075061 70.7 48.6  12.4 186.0 -0.088 -0.052   0.376 0.378 0.364 0.360 0.367 0.339  0.0031  2
    080307 075061 70.7 41.9   9.4 186.0 -0.106 -0.060   0.360 0.348 0.341 0.353 0.361 0.325  0.0067  2
    080307 075061 70.7 34.0   4.6 186.0 -0.127 -0.069   0.347 0.342 0.341 0.352 0.352 0.316  0.0065  2
    ...
```

Each file starts with 3 header lines. The header provides the pixel location in latitude and longitude, the IGBP class number, the NDVI, the number of valid satellite overpasses, the number of valid observations (each overpass provides up to 16 different directional measurements) and the fraction of the dominant surface type within the POLDER pixel (about 6x6 km$^2$).

The format of the second line of the header that contains the numerical values is

fmtHead='(F8.2,3x,F8.2,5x,I3,4x,F7.2,5x,I3,5x,I4,6x,I3)'

The header is followed by the measurements. Each line corresponds to one directional observation. The "no data" value is -9.99. The format of the different columns is:

fmtLine='(I6,x,I6,2F5.1,2F6.1,2F7.3,2x,6F6.3,F8.4,I3)'

Each line contains the following information:

**yymmdd** is the date of observation.




**Orbit** provides the PARASOL orbit "cccooo" where "ccc" is the PARASOL cycle number (1<ccc<999) and "ooo" is the orbit number (1<ooo<233).

**SZA** is the sun zenith angle in degrees.

**VZA** is the view zenith angle in degrees.

**RelAzi** is the relative azimuth in degrees.

**AziS** is the sun azimuth with respect to the North direction, in degrees.

**DVzC** and **DVzS** can be used for slight corrections of the view geometry. Indeed, POLDER spectral measurements are not simultaneous so that each channel is acquired with a slightly different viewing geometry. The view angles that are given are for the 670 nm band. The view geometry

for the other channels can vary by a few tenths of degrees. For applications that require a higher accuracy, these parameters allow the correction that is described in appendix B.

**RXXX** are the surface reflectances at 490, 565, 670, 765, 865, and 1020 nm.

**Rp865** is the surface polarized reflectance at 865 nm. Positive values indicate a polarisation perpendicular to the plane of scattering. Negative values indicate a polarisation parallel to the

plane of scattering. Further explanation is section 2.2.

**Aero** is a non-quantitative indication of the aerosol load retrieved from POLDER measurements. 0 is for minimal aerosol load, whereas 15 is for a high aerosol load.

## 7.2   Appendix B: Compute the exact view direction for all channels

With the POLDER/Parasol imaging concept, the 15 spectral/polarized measurements are acquired

sequentially. Therefore, a given surface target is observed, for the various spectral bands, with slightly different viewing angles. The differences are small, but can be significant for some applications that need a very high angular accuracy, such as the analysis of the Hot-Spot directional signature.

The view zenith angle ($\theta_0$) and relative azimuth ($\phi_0$) that are given in the BRDF database are for the central filter, i.e. 670P2. The two parameters $DVzC=\Delta[\theta_v \cos(\phi)]$ and $DVzS=\Delta[\theta_v \sin(\phi)]$, which are

given for each viewing direction in the data file, are necessary to derive these angles for other spectral bands $\theta_j$ and $\varphi_j$. The formulae are as follows:





$$\theta_j = \sqrt{\left(\theta_0 cos\phi_0 + X_j DVzC\right)^2 + \left(\theta_0 sin\phi_0 + X_j DVzS\right)^2}$$

$$\phi_j = arctan\left(\frac{\theta_0 sin\phi_0 + X_j DVzS}{\theta_0 cos\phi_0 + X_j DVzC}\right)$$

If $\theta_0 sin\phi_0 + X_j DVzS < 0$ then $\phi_j = \phi_j + 180°$

where $Xj$ is given in the table below:

| $Xj=$ | -6 | -4 | -3 | -2 | 0 | 2 | 3 | 4 | 6 |
|---|---|---|---|---|---|---|---|---|---|
| Channel | 490P | 443 | 1020 | 565 | 670 | 763 | 765 | 910 | 865 |

Note: This formulation is based on the simple principle that the 15 measurements are acquired with roughly equal spacing and on a straight line in an angular system of orthogonal axes ($\theta\ cos\phi,\ \theta\ sin\phi$).




# Tables and Figures

| Central Wavelength | 443 | 490 | 565 | 670 | 763 | 765 | 865 | 910 | 1020 |
|---|---|---|---|---|---|---|---|---|---|
| Polarization | N | Y | N | Y | N | N | Y | N | N |
| In database | N | Y | Y | Y | N | Y | Y | N | Y |

**Table 1:** POLDER/Parasol spectral bands. The second line indicates the polarized channels, whereas the third lines indicates the bands that are included in the BRDF/BPDF database.

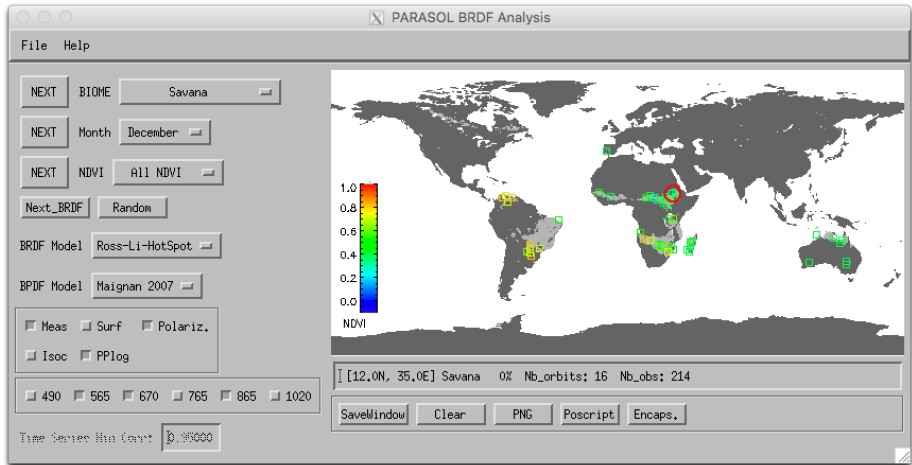

**Figure 1:** *Main Command Window* (*MCW*) of the visu_BRDF analysis tool. Various buttons and drop-down menus permit the selection of a given target and various display options as described in the text.



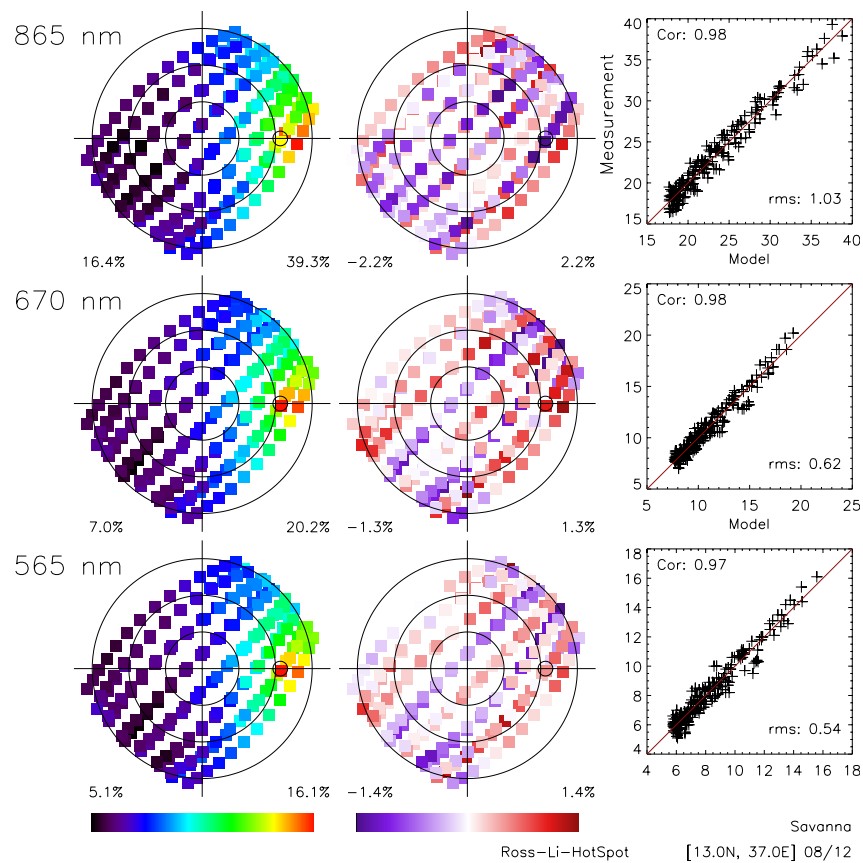

**Figure 2:** An example of the "*Target BRDF*" window content. The left column shows the measurements, the middle column show the difference between the measurements and the modelling, and the left column is a measurement-model scatter plot. Three channels are shown which can be selected among 6 (490, 565, 670, 765, 865 and 1020 nm). Note that the scales are different between channels.




**Figure 3:** An example of the Principal plane (left) and perpendicular plane (right) window contents. The measurements are shown as triangles and the model prediction is indicated as a line. Three channels are shown which can be selected among 6. The Y-scale can be either linear of logarithmic depending on the *MCW* setting (**Figure 1**).

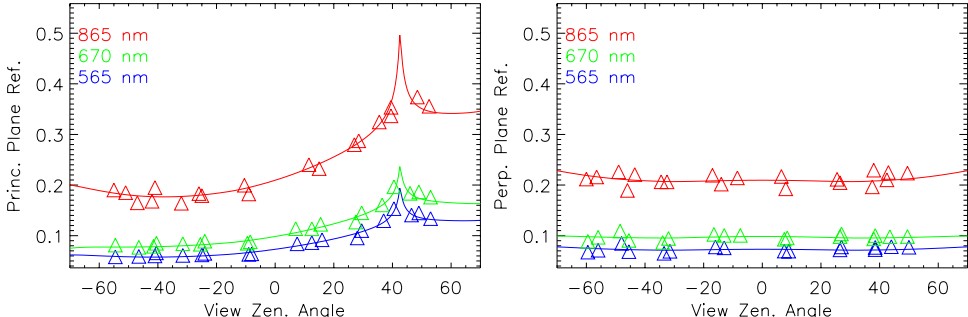

**Figure 4:** An example of the "*Target BPDF*" window contents. The presentation is similar as for **Figure 2**, but only the polarized
10   reflectance at 865 nm is shown.





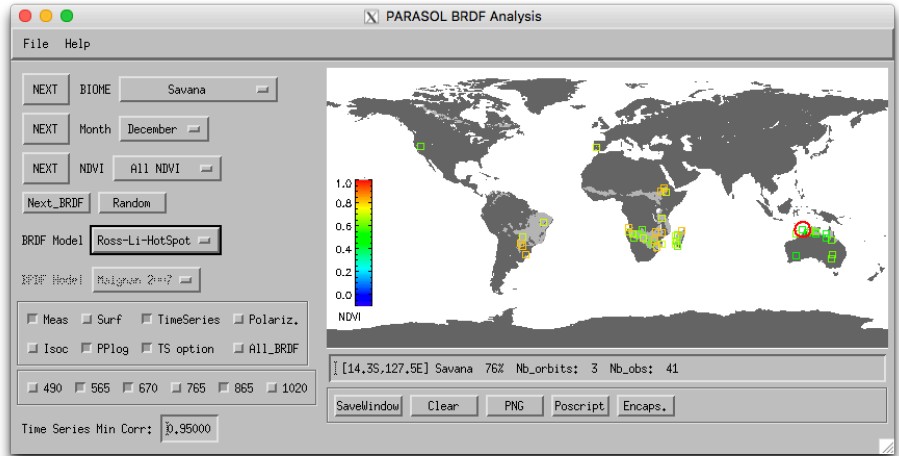

**Figure 5:** *Main Command Window* (*MCW*) when the yearly database is used as input. There are additional options (from the *MCW* of **Figure 1**) to display time series of various parameters

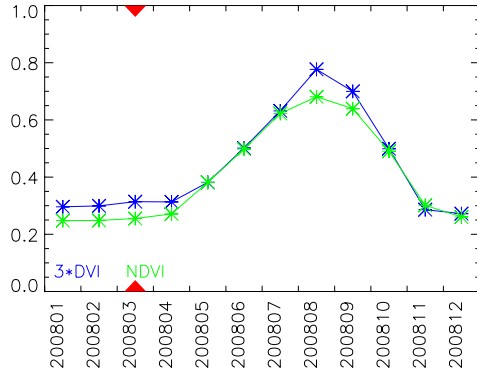

**Figure 6:** An example of the *NDVI* window content. It shows the time series of the NDVI and the DVI. The red triangles indicate the month that is displayed on other windows (Figures 2-4).



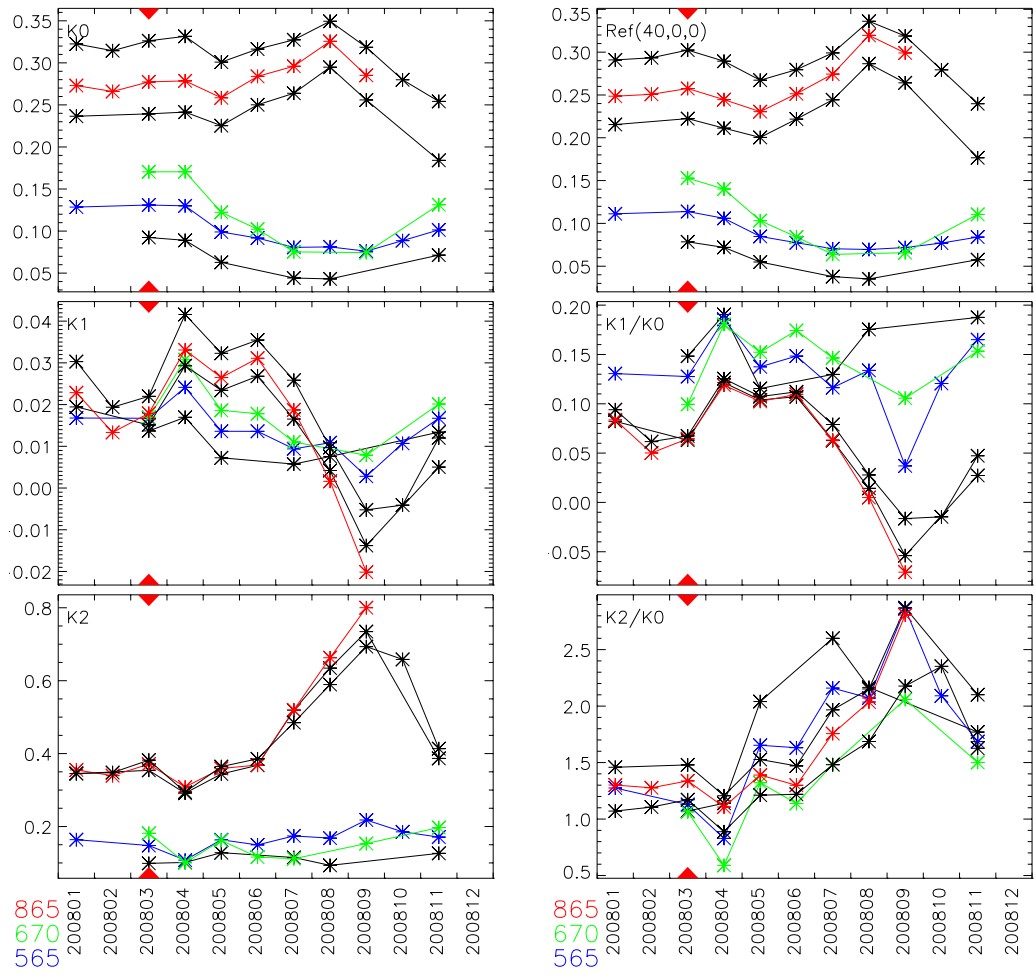

**Figure 7:** Example of the "*Time Series*" window content. Depending on a setting of the *MCW*, this window shows either the monthly time series of the BRDF model parameters (left), or a combination of the same. For the later option, the top figure is the reconstructed reflectance for a reference observation geometry. On these figures, the coloured line/symbols are for the 3 selected bands while the same parameters for the other bands are also shown in black.





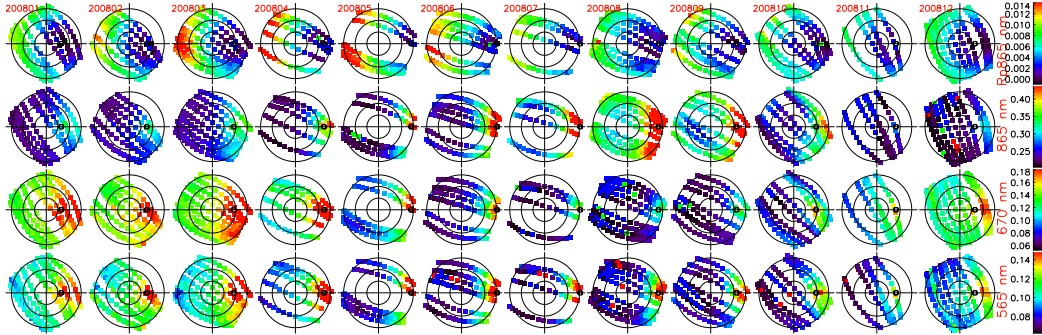

**Figure 8: An example of the content of the "*All BRDF/BPDF*" window. This window shows, for a given target, the measurements for all 12 months in the database. The top line is the BPDF at 865nm while the other three lines are for the reflectances for the three selected bands among 6.**