# Peer review of "A BRDF-BPDF database for the analysis of Earth targets reflectances"

_Earth System Science Data, 2016_

## Referee Comment (RC1) · Anonymous Referee #1 · 18 Oct 2016

General comments

This paper presents a new BRDF/BPDF database derived from 1 year of POLDER data, categorised according to land cover class. The database is of interest to people looking at the variability of reflectance and polarization as a function of land cover type, and over time. The paper is generally clear, well-written and presents useful data, as well as a tool to view them.

I think the data products are potentially useful and described well. I think the issue of spatial scale needs to be discussed more clearly (see page 16, 22: Yes - it may be argued!) and caveats given at the outset in terms of the limitations of the original data and the resulting database. It's not a problem per se BUT you ought to be clear that this is perhaps the single biggest limitation of the data and if someone is interested in

finer scale variability, then this database will be of little use to them. In addition, there is a lack of consideration from the figures (eg 6 and 7) of the spatial variability. Could this be shown as error bars representing the average of the underlying pixels making up each point?

page 7, line 7: where does the threshold of 75% come from, to determine the dominance of one cover type? This could potentially have some impact on the resulting database in terms of the number of polder pixels of each class that are processed. It might be useful to give an idea of this eg by giving some figures on how many pixels of each class are kept at 70, 75, 80 and 90% level. Otherwise, wouldn't this affect the score[p,m]? The same question arises over the arbitrary increase in score of 20% for observations with small phase angle. Why 20% and why 1 degree? These at least need some explanation in terms of how different choices might affect the scores and database size/quality.

Lastly, I am less convinced about the amount of time/space given over to describing the tool. Fine, it seems useful for viewing, but most people who use the database will surely want to read the data directly using their own matlab, python etc code. And so a tool developed using a proprietary language (IDL), even with the free runtime idl vm, is not really particularly open/useful, but takes up a lot of space in the manuscript. I would suggest putting all that material (description of tool and user instructions) in an appendix, or remove it entirely and just refer to a web link for download and documentation. Then the authors could just mention the figures were prepared using that tool.

Specific comments

page 1 line 12: Note that albedo is also difficult to measure at is not an intrinsic surface property; it depends on the direct/diffuse ratio of incoming radiation and hence is a function of atmospheric state. 1, 22: optical. The shadow is not an issue for spaceborne measurements! 5, 10: what do you mean by best year? Total number of

cloud-free observations? Or according to criteria in section 2.3 on page 6? p 15, 17: perhaps provide the rationale behind showing the ratio of parameters here, rather than in different places below. It;s not immediately clear why this is useful. Some explanations are given further down, but this could be summarised more clearly here i.e. k1/k0 should indicate ..... k2/k0 should indicate ..... But also won't this depend on which model is chosen, as k0, k1 and k2 can have fairly different interpretation depending on model choice? You allude to this but are not clear. For example the k2 parameter from RossThick is very specifically the volume scattering component, assuming an LAI » 2. So it's things like that which could be clearer - maybe put in a table? Figure 8: I'm not convinced by the usefulness of this. Too much information on one plot with points being much too small. Separate figures for each line with larger plots in each case might help (eg down the page, or change page orientation to landscape).

Grammatical/technical

page 5, line 15: comma after sensitivity - otherwise this sentence is very long and a bit hard to follow. 5, 16: These developments 5, 19: why capitals for Top and Reflectances? 6, 24: typo after reference 6, 25: deserts show ... wetlands sometimes .... 7, 1: why ellipsis? 8, 2: ensures 8, 21: Change title of section 3 to something like: Analysis of the database features using the visu_brdf analysis tool 12, 3: reflectances, or reflectance values (plural). And you say typical - you probably need to clarify that in terms of these characteristic properties of dense or sparse vegetation, as it can vary a lot otherwise. 13, 10: small but IS necessary 13, 18: what does "properly" mean? 15, 1: Section 3.7.1 - define NDVI and DVI properly here, to avoid any confusion. 15, 5: fit to the measurements

---

## Referee Comment (RC2) · Anonymous Referee #2 · 27 Oct 2016

The POLDER/PARASOL database constitutes an unrivaled historical record for understanding polarization effects of surface directional reflectance. We know that polarization can have a very large effect on reflectance, particularly in the case of specular reflectance, so this database is a great resource for understanding polarization effects on BRDF. I recommend publication in ESSD but think that some technical points and discussions should be updated first to enhance the clarity and technical correctness of the paper.

General comments: 1. According to Nicodemus et al. (1977): the BRDF is a derivative, a distribution function, relating the irradiance incident from one given direction to its contribution to the reflected radiance in another direction. So the question I now have is how do we define BPDF (bidirectional polarization distribution function)? Does BPDF follow the same nomenclature as BRDF without any modification? The

authors need to show how the two are related. 2. This publication is too similar to published posters shown below and raises the questions whether there is any need to publish the material in ESSD: a. Bréon, F.M., E. Fédèle, F. Maignan, and R. Lacaze, A database of directional reflectance signature (IGBP) with an analysis tool, A-Train Symposium, Lille, 22-25 October 2007 – http://postel.obs-mip.fr/IMG/pdf/Poster_BRDF_PARASOL_ColloqueAtrain2007_IGBP.pdf & b. Bréon, F.M., E. Fédèle, F. Maignan, and R. Lacaze, A database of directional reflectance signature (GLC2000) with an analysis tool, A-Train Symposium, Lille, 22-25 October 2007 – http://postel.obs-mip.fr/IMG/pdf/Poster_BRDF_PARASOL_ColloqueAtrain2007_GLC2000.pdf

3. Some of the comments in the code (visu_brdf.pro) are in French. Why not include English translation, where applicable. 4. Atmospheric correction should be clearly described, how in particular the radiative transfer problem is modeled in terms of surface BRDF and how polarization is taken into account. 5. Also, describe how E0 is derived in Eq. 1 and 2.

Other minor technical corrections 1. there are negative values in the database, which need to be explained. 2. Pg. 2, line 8, clarify with some examples the statement "Many land surface characteristics can be inferred from the spectral signature of their albedo" 3. Pg. 2, line #21, change "optic" to "optical" 4. Pg. 2, line #22, change "properly at" to "properly as" 5. Pg. 3, line #2, clarify the statement "the azimuths are only significant by their difference." 6. Pg. 3, line #4, add "respectfully" after "angles" 7. Pg. 3, line #26, what does "confrontation to analytical models" mean? 8. Pg. 4, line #10, change "name" to "named" 9. Pg. 4, line #17, change "by step" to "by a step" 10. Pg. 5, line #10, change "term" to "terms

---

## Author Comment (AC1) · 10 Dec 2016

Answer to reviews

In the following, the reviewer comments are in italic, while our answers are in plain text.

**Anonymous Referee #1**

**General comments**

*This paper presents a new BRDF/BPDF database derived from 1 year of POLDER data, categorised according to land cover class. The database is of interest to people looking at the variability of reflectance and polarization as a function of land cover type, and over time. The paper is generally clear, well-written and presents useful data, as well as a tool to view them.*

We thank the reviewer for this summary

*I think the data products are potentially useful and described well. I think the issue of spatial scale needs to be discussed more clearly (see page 16, 22: Yes - it may be argued!) and caveats given at the outset in terms of the limitations of the original data and the resulting database. It's not a problem per se BUT you ought to be clear that this is perhaps the single biggest limitation of the data and if someone is interested in finer scale variability, then this database will be of little use to them. In addition, there is a lack of consideration from the figures (eg 6 and 7) of the spatial variability. Could this be shown as error bars representing the average of the underlying pixels making up each point?*

Note that we certainly did not try to hide this limitation of the database, as this limitation is explicitly stated in the paper. We agree with the reviewer that it could be elaborated, and we shall add a discussion on the spatial resolution. In short, we have targeted homogeneous areas, so that no scale effects are expected, but high resolution observations are required to validate this statement. Such analysis and validation is clearly out of scope of this paper. Concerning Figure 6 and 7, we do not understand the reviewer point. These figures (as for the others) are for a single pixel. We have no information on the spatial variability that could be shown on these figures.

*page 7, line 7: where does the threshold of 75% come from, to determine the dominance of one cover type? This could potentially have some impact on the resulting database in terms of the number of polder pixels of each class that are processed. It might be useful to give an idea of this eg by giving some figures on how many pixels of each class are kept at 70, 75, 80 and 90% level.*

We acknowledge that the 75% threshold is somewhat empirical. The selection of targets with 100% homogeneity led to empty classes for some surface types. We then decreased the threshold until there was a sufficient number of valid targets in the database.

*Otherwise, wouldn't this affect the score[p,m]? The same question arises over the arbitrary increase in score of 20% for observations with small phase angle. Why 20% and why 1 degree? These at least need some explanation in terms of how different choices might affect the scores and database size/quality.*

1° is the typical angular width of the Hot Spot, which led to the threshold selection. The 20% score increase for the targets with such angular observation is more empirical and based on our observation of the typical scores and their variability. We shall add a few sentences for justification.

*Lastly, I am less convinced about the amount of time/space given over to describing the tool. Fine, it seems useful for viewing, but most people who use the database will surely want to read the data directly using their own matlab, python etc code. And so a tool developed using a*

*proprietary language (IDL), even with the free runtime idl-vm, is not really particularly open/useful, but takes up a lot of space in the manuscript. I would suggest putting all that material (description of tool and user instructions) in an appendix, or remove it entirely and just refer to a web link for download and documentation. Then the authors could just mention the figures were prepared using that tool.*

We tend to disagree. Although the users experienced with BRDF may want to analyze the database with their own tools, there is also a wide audience that knows little about directional effects, their variability, BRDF modeling and the ability for the models to reproduce the observed directional signatures. The tool is designed for these potential users who can then get a quick and easy feeling about these questions. In addition, the paper uses the tool description to show the content of the database and how it can be used. We thus feel it is an important feature of the paper.

**Specific comments**

*page 1 line 12: Note that albedo is also difficult to measure at is not an intrinsic surface property; it depends on the direct/diffuse ratio of incoming radiation and hence is a function of atmospheric state.*

Agreed

*1, 22: optical. The shadow is not an issue for spaceborne measurements!*

We fully disagree. Shadows are present in the spaceborne measurements. In fact, the Hot Spot effect can be explained by the fact that foliage element hide their own shadows (ie it is the only direction for which the spaceborne instrument does not see shadows, which explains the bright reflectance

*5, 10: what do you mean by best year? Total number of cloud-free observations? Or according to criteria in section 2.3 on page 6?*

The paper states that it is the best in term of data acquisition. The POLDER/Parasol instrument suffered from several period in safe mode due to malfunction of the stellar sensor. It was seldom the case during 2008. We shall make that clear in the manuscript.

*p 15, 17: perhaps provide the rationale behind showing the ratio of parameters here, rather than in different places below. It;s not immediately clear why this is useful. Some explanations are given further down, but this could be summarised more clearly here i.e. k1/k0 should indicate ..... k2/k0 should indicate ..... But also won't this depend on which model is chosen, as k0, k1 and k2 can have fairly different interpretation depending on model choice? You allude to this but are not clear. For example the k2 parameter from RossThick is very specifically the volume scattering component, assuming an LAI >> 2. So it's things like that which could be clearer - maybe put in a table?*

We agreed that we shall provide some explanation of the rationale behind the use of ratio parameters. These ratios expresses the BRDF amplitude in relative units. In the single scattering approximation (which is almost valid in the visible as the reflectance is rather low), the directional effects are generated by the target architecture while its overall amplitude depends on the reflectance of the individual elements. As a consequence, the BRDF can be expressed as the product of a normalized reflectance and a normalized shape. The numerical expression of this shape is based on the parameter ratios. We shall make this clearer as rightly recommended by the reviewer.

*Figure 8: I'm not convinced by the usefulness of this. Too much information on one plot with points being much too small. Separate figures for each line with larger plots in each case might help (eg down the page, or change page orientation to landscape).*

We tend to disagree about the usefulness of such figure. It certainly does contain a lot of information, but that is what makes it useful. Indeed, the point size may be an issue on a small screen or with the page dimension limitation. We shall put it in landscape mode in the revised version.

**Grammatical/technical**

*page 5, line 15: comma after sensitivity - otherwise this sentence is very long and a bit hard to follow.*
*5, 16: These developments*
*5, 19: why capitals for Top and Reflectances?*
*6, 24: typo after reference*
*6, 25: deserts show ... wetlands sometimes....*
*7, 1: why ellipsis?*
*8, 2: ensures*
*8, 21: Change title of section 3 to something like: Analysis of the database features using the visu_brdf analysis tool*
*12, 3: reflectances, or reflectance values (plural). And you say typical - you probably need to clarify that in terms of these characteristic properties of dense or sparse vegetation, as it can vary a lot otherwise.*
*13, 10: small but IS necessary*
*13, 18: what does "properly" mean?*
*15, 1: Section 3.7.1 - define NDVI and DVI properly here, to avoid any confusion.*
*15, 5: fit to the measurements*

We are very much grateful for these technical/grammatical corrections that we shall implement.

**Anonymous Referee #2**

*The POLDER/PARASOL database constitutes an unrivaled historical record for understanding polarization effects of surface directional reflectance. We know that polarization can have a very large effect on reflectance, particularly in the case of specular reflectance, so this database is a great resource for understanding polarization effects on BRDF. I recommend publication in ESSD but think that some technical points and discussions should be updated first to enhance the clarity and technical correctness of the paper.*

We thank the reviewer for this summary and positive comment

**General comments:**

*1. According to Nicodemus et al. (1977): the BRDF is a derivative, a distribution function, relating the irradiance incident from one given direction to its contribution to the reflected radiance in another direction. So the question I now have is how do we define BPDF (bidirectional polarization distribution function)? Does BPDF follow the same nomenclature as BRDF without any modification? The authors need to show how the two are related.*

Indeed, we define BPDF very similarly as the BRDF. Whereas the BRDF is a distribution function of the reflected radiance (R, derived from I as in eq. 2), the BPDF is a distribution function of the polarized reflected radiance (Rp, derived from Q as in eq. 1). We will make this point clear in the revised version of the manuscript.

*2. This publication is too similar to published posters shown below and raises the questions whether there is any need to publish the material in ESSD:*

*a. Bréon, F.M., E. Fédèle, F. Maignan, and R. Lacaze, A database of directional reflectance signature (IGBP) with an analysis tool, A-Train Symposium, Lille, 22-25 October 2007 – http://postel.obs-mip.fr/IMG/pdf/Poster_BRDF_PARASOL_ColloqueAtrain2007_IGBP.pdf*

*& b. Bréon, F.M., E. Fédèle, F. Maignan, and R. Lacaze, A database of directional reflectance signature (GLC2000) with an analysis tool, A-Train Symposium, Lille, 22-25 October 2007 http://postel.obs-mip.fr/IMG/pdf/Poster_BRDF_PARASOL_ColloqueAtrain2007_GLC2000.pdf*

These two published material are conference posters. It is a standard and accepted procedure to publish papers that use materials from conference posters. In addition, the database and the analysis tool benefit from several improvements since these conferences, including full re-calibration, addition of the polarized component, and new graphical options.

*3. Some of the comments in the code (visu_brdf.pro) are in French. Why not include English translation, where applicable.*

We agree that this is a shortcoming, impractical to some users. We shall make a full translation to English of the comments embedded in the IDL code and make this new version available, together with the new version of the paper.

*4. Atmospheric correction should be clearly described, how in particular the radiative transfer problem is modeled in terms of surface BRDF and how polarization is taken into account. 5. Also, describe how E0 is derived in Eq. 1 and 2.*

The atmospheric correction and the associated inversion of the aerosol load is rather difficult and cannot be described in a few sentences. This is why we rather make a reference to a published paper. As for E0, it is clearly described as the TOA solar irradiance.

**Other minor technical corrections**

*1. there are negative values in the database, which need to be explained.*

Some negative values correspond to polarized reflectance when the polarization plane is parallel to the plane of scattering as explained towards the end of section 2.2. The other (-9.990) are fill values. We will make this last point clear in the paper appendix.

*2. Pg. 2, line 8, clarify with some examples the statement "Many land surface characteristics can be inferred from the spectral signature of their albedo"*

We shall provide references for this particular point

The other suggested correction below are all accounted for

*3. Pg. 2, line #21, change "optic" to "optical"*

*4. Pg. 2, line #22, change "properly at" to "properly as"*

*5. Pg. 3, line #2, clarify the statement "the azimuths are only significant by their difference." 6. Pg. 3, line #4, add "respectfully" after "angles"*

*7. Pg. 3, line #26, what does "confrontation to analytical models" mean?*

*8. Pg. 4, line #10, change "name" to "named"*

*9. Pg. 4, line #17, change "by step" to "by a step"*

*10. Pg. 5, line #10, change "term" to "terms*